# Stereotactic Body Radiation Therapy (SBRT) Plus Immune Checkpoint Inhibitors (ICI) in Hepatocellular Carcinoma and Cholangiocarcinoma

**DOI:** 10.3390/cancers15010050

**Published:** 2022-12-22

**Authors:** Joanna Jiang, Dayssy Alexandra Diaz, Surya Pratik Nuguru, Arjun Mittra, Ashish Manne

**Affiliations:** 1Department of Internal Medicine, The Ohio State University College of Medicine, Columbus, OH 43210, USA; 2Department of Radiation Oncology, Ohio State University James Comprehensive Cancer Center, Columbus, OH 43210, USA; 3School of Medicine, Kamineni Academy of Medical Sciences and Research Center, Hyderabad 500012, India; 4Department of Internal Medicine, Division of Medical Oncology at the Arthur G. James Cancer Hospital and Richard J. Solove Research Institute, The Ohio State University Comprehensive Cancer Center, Columbus, OH 43210, USA

**Keywords:** hepatocellular cancer, cholangiocarcinoma, immunotherapy, SBRT, radiation, checkpoint, liver cancer, radiation, liver cancers, combination therapy

## Abstract

**Simple Summary:**

Cancers arising from the liver and the biliary tract are aggressive and have limited treatment options. The recent success of immunotherapy where in patient’s immune system is activated to fight the tumor is encouraging, but only a fraction of patients with liver cancer remain eligible for this treatment. There is a good pre-clinical evidence (from animal studies) that combination with high-dose focused radiation or stereotactic body radiation (SBRT) makes immunotherapy more effective in these cancers. In this review the available evidence for such combination in treating biliary tract and liver cancers was explored in depth. The preliminary evidence suggests that combining SBRT and immunotherapy is safe and there is a need for large scale trials to investigate its efficacy.

**Abstract:**

The combination of stereotactic body radiation therapy (SBRT) plus immune checkpoint inhibitors (ICI) must be explored to treat advanced primary liver tumors such as hepatocellular carcinoma (HCC) and cholangiocarcinoma (CCA). Limited retrospective reviews and case reports/series suggest this combination can be effective and safe in both cancer types. With ICIs moving into the first line (IMbrave 150, HIMALAYA, and TOPAZ-1) to manage these cancers, identifying a suitable population for this approach is challenging. Patients with macrovascular invasion (MVI)-positive HCC (especially if larger veins are involved) or recurrent HCCs post-locoregional therapies (such as transarterial radioembolization (TARE), transarterial chemoembolization (TACE), or ablation), as well as those ineligible for bevacizumab or tyrosine kinase inhibitors (TKIs), should be the focus of exploring this combination in HCC. Unresectable or oligometastatic CCA patients who cannot tolerate gemcitabine/cisplatin (GC) or those who progressed on GC without durvalumab and do not have targetable mutations could also be considered for this approach. In both HCC and CCA disease groups, SBRT plus ICI can be examined post-ICI as these two modalities act synergistically to enhance anti-tumor activity (based on pre-clinical studies). Large-scale randomized trials are needed to identify the subsets of primary liver cancers suitable for this approach and to clearly define its clinical benefit.

## 1. Introduction

Hepatocellular carcinoma (HCC) and cholangiocarcinoma (CCA) are the two most common primary liver tumors, accounting for over 80% of all hepatobiliary malignancies [1]. Both cancers have high mortality rates, and their rising incidence in the last five years is alarming [2,3,4,5]. Systemic options to treat these cancers are limited, and there is a need to improve the available modalities to manage them effectively.

Multiple liver disease groups have proposed guidelines for managing HCC that aid in risk stratification and treatment selection [6,7,8,9,10,11,12]. The locoregional therapy (LRT) options range from and ablation in early-stage tumors to transarterial chemoembolization (TACE), transarterial radioembolization (TARE), and stereotactic body radiation therapy (SBRT) in intermediate-stage tumors and selective advanced-stage HCC tumors (aHCC) [13,14,15]. Severe adverse events such as duodenal ulcer, pneumonitis, and gastrointestinal bleeding were associated with traditional (external beam) radiation but not with SBRT according to the metanalysis that compared various radiation modalities in HCC patients with portal vein tumor thrombus (PVTT) [16]. While lymphocytopenia is the most common adverse event with traditional radiation, severe thrombocytopenia was the most frequently reported adverse event (15%). Resection is offered to eligible early-stage tumors whenever it is feasible. Systemic therapy typically reserved for aHCC can be divided into three broad categories: immune-checkpoint inhibitors (ICIs), tyrosine kinase inhibitors (TKIs), and vascular endothelial growth factor inhibitors (VEGFis) [14]. Until mid-2020, sorafenib (a TKI) was offered as a first-line treatment, but now atezolizumab/bevacizumab can be offered if there are no contraindications for ICI and VEGFi use [17]. The results of HIMALAYA trial (durvalumab/tremelimumab vs. sorafenib) are encouraging and may be another firstline option in the future [18]. Single-agent ICIs, such as pembrolizumab and nivolumab, are reserved for patients who progressed on or are ineligible for sorafenib [19,20,21,22,23].

The gemcitabine/cisplatin (GC) combination was the standard of care to treat advanced-stage CCA (aCCA) until the TOPAZ-1 trial showed the benefit of adding durvalumab to GC [24,25]. Microsatellite instability-high (MSI-H) and tumor mutational burden-high (≥10 mutations/megabase) CCA patients can be treated with pembrolizumab [26,27]. Nivolumab was reserved for CCAs refractory to standard systemic options before durvalumab became the standard of care [28]. Unlike in HCC, SBRT use in CCA is not the standard of care. Adjuvant external radiation is offered for extrahepatic CCA (and gallbladder cancers) with positive margins [29]. There is no level-I evidence supporting the use of radiation therapy as the first-line treatment in aCCA.

Combining LRT with systemic therapy is not typical for HCC and CCA. Major trials on ICI in HCC did not clearly mention the use of SBRT (or LRT) in inclusion/exclusion criteria; however, some required 4 weeks between regional therapy and initiation of ICI (further information regarding eligibility criteria of all major trials concerning the timing of LRT and liver function in Appendix A). While such combinations with TARE are being evaluated in multiple trials for various HCC and CCA patient populations, the success of combination SBRT and ICI in other solid tumors such as non-small-cell lung cancer (NSCLC) has prompted interest in exploring such an approach to primary liver tumors [30,31,32,33]. SBRT is a form of external beam radiotherapy wherein a single or limited number (3 to 6) of high-dose radiation fractions are delivered to a targeted area [34]. Earlier attempts to add traditional chemotherapy (for CCA), TKI (for HCC), and VEGFi (for all tumors) to SBRT proved to be toxic and is discouraged in the current guidelines [35,36,37,38,39,40,41].

## 2. The Rationale of Adding ICI to SBRT

Immunotherapy refers to modalities that modify the immune system to achieve desirable therapeutic benefits [42]. In current oncology practice, ICI is a widely used form of immunotherapy. Oncolytic virus therapies, vaccines, cytokines, and adoptive cell transfer are other kinds of immunotherapies [43]. Pre-clinical evidence suggests that immunotherapy works synergistically with radiation to enhance tumor-specific immune responses [44,45,46] (see Figure 1).

Solid tumors can avoid anti-tumor immunity by downregulating antigen presentation and CD8+ cytotoxic T-cell activity [47]. Immunotherapy counteracts this by either stimulating activating regulators of T-cell function (OX40) or inhibiting negative regulators such as ICI, including the programmed death 1 receptor (PD-1), programmed cell death ligand 1 (PD-L1), and cytotoxic T-lymphocyte-associated antigen 4 (CTLA-4) [48,49]. ICIs that target PD-1 increase polarization of tumor-associated macrophages and activate cytotoxic T cells. Anti-PD-L1 inhibits tumor immune evasion. Anti-CTLA4 also inhibits immune evasion, and increases interaction between dendritic cells and T cells. Combining ICIs with VEGFis or TKIs enhance these effects by promoting dendritic cell and T lymphocyte activity, and inhibiting regulatory T cells and myeloid-derived suppressor cells. This results in increased anti-tumor inflammation and a more durable response to ICIs [14].

Immunotherapy has been shown to increase antigen presentation and anti-tumor inflammation in animal models of solid tumors, including glioblastoma, lymphoma, and breast and lung cancers. In one orthotopic murine HCC model, response to stereotactic radiation was augmented by concurrent treatment with PD-1 antibodies [49]. However, that response was transient as the combination also induced T helper 2-suppressive responses which in-turn promoted immune resistance. These studies gave reliable evidence on the benefit with combining SBRT and ICI but also caution the possible the immune resistance it can promote.

The optimal timing of ICI relative to radiation therapy (concurrent, before, or after) is undetermined and has varied across animal studies [50,51,52,53,54]. Two studies found that a CTLA-4 blockade combined with radiation therapy appears to enhance T-cell response within tumors and improve survival in mouse models: one study administered anti-CTLA-4 before radiation therapy, while the other administered treatments in reverse order. Another pre-clinical study by Young et al. demonstrated enhanced efficacy when anti-CTLA-4 was given prior to radiation therapy, likely due to the depletion of regulatory T cells. In contrast, anti-OX40 was most effective when given one day after radiation to enhance antigen presentation. Conflicting evidence exists regarding optimal timing for a PD-L1 blockade, with one study supporting radiation therapy followed by immunotherapy and another finding that only concomitant rather than sequential treatment improved survival [50,52].

In summary, anti-tumor responses can be enhanced by adding immunotherapy to radiation (concurrent vs. sequential), but the response is short-lived. Continuing immunotherapy post-radiation could make that response durable. In clinical practice, this was practically translated to SBRT plus ICI in NSCLC where durvalumab was continued after SBRT for locally advanced tumors.

## 3. SBRT Plus ICI in Hepatocellular Carcinoma

One case series and a terminated phase I trial gave insight into the SBRT plus ICI combination in HCC [55,56]. The baseline characteristics and significant findings are summarized in Table 1 (below).

In the case series with five uHCCs, notably, the response rates differed with the criteria used to assess the response. According to Response Evaluation Criteria in Solid Tumors (RECIST) criteria, all patients had partial response or PR (ranging from 30–84%). On the other hand, two patients had a complete response (CR), and three had PR when assessed by modified RECIST criteria. In the phase I trial closed due to slow accrual, uHCC patients received nivolumab or ipilimumab/nivolumab. This trial evaluated the response by immune-related RECIST (iRECIST) [59]. None of the participants had CR, and the group that received dual ICI therapy responded (details in Table 1). Serious adverse events were not frequent. One patient developed grade ≥3 toxicity (both pneumonitis and skin), and two (out of 14) patients had dose-limiting toxicity (hepatotoxicity). These two studies suggest that this approach can work and is tolerable in HCC, even in patients with previous LRTs such as TACE.

In the last 3 to 4 years, numerous phase II and III trials looking at SBRT plus ICI in HCC were initiated (Table 2). The timing of SBRT and the initiation of ICI varies. In the trials with a sequential approach (SBRT followed by ICI or vice versa), 2 of 3 trials were specifically for patients with MVI (NCT04167293; NCT04169399). In one trial, uHCC patients were first treated with TACE and SBRT; ICI (durvalumab every four weeks and one dose of tremelimumab) was administered later with a goal of downstaging for surgical resection (NCT04988945). A group at Massachusetts General Hospital is currently recruiting patients with resectable HCC (NCT04857684). Patients will be treated with neoadjuvant SBRT combined with atezolizumab/bevacizumab, and treatment will be monitored for safety and tolerability. Two trials are currently investigating outcomes of ICI prior to SBRT.

Six trials will use concurrent treatment with radiation ICI. One will administer durvalumab (anti-PD-1), tremelimumab (anti-CTLA4), and radiation therapy (not necessarily SBRT) during the second cycle of ICIs for approximately 70 participants with HCC and biliary tract cancers (NCT03482102). The other five trials, currently underway e in Canada and China, will use anti-PD-1 ICIs such as pembrolizumab (NCT03316872), tislelizumab (NCT05185531), carelizumab (NCT04193696), and sintilimab (NCT04547452, NCT03857815). The pembrolizumab and careluzumab trials will focus on advanced HCC that has failed previous therapy, while the tislelizumab trial is for neoadjuvant therapy with an endpoint of delay in surgery. The sintilimab trials will recruit advanced HCC with metastatic disease, and SBRT can be given to liver or metastatic sites (lung or any metastatic lesion), and early results of one trial (NCT03857815) were presented at ASCO this year [60]. At data cut-off, 25 out of 30 planned patients were recruited (median follow up of 17 months), and confirmed ORR was 96% (18 complete response and 6 partial response) with 100% local control rate. Serious adverse events (SAE) reported in 3 patients were myocarditis, viral hepatitis, and gastrointestinal bleed (GIB).

### Identifying Ideal HCC Patients for SBRT Plus ICI

According to the American Association for the Study of Liver Diseases (AASLD) guidelines, intermediate-stage or Barcelona clinic liver cancer (BCLC)-B or BCLC-B HCC can be treated with SBRT [6]. To our knowledge, there are no reliable predictors for a robust response for SBRT plus ICI combination. Selecting ideal patients for this modality is key as demonstrated by Bujold et al. [61]. The authors presented the data on SBRT in HCCs unsuitable for traditional LRTs (*n* = 102) from sequential phase I and phase II trials [61]. The eligibility criteria were similar for both trials, except in the phase II trial, there was a limit on the number (≤5) and maximal dimension (15 cm) of the liver lesions. The outcome in patients with the second trial (with restrictions on size and number of liver lesions) was better than the first, suggesting the role of tumor burden on outcomes (hazards ratio or HR of 0.49, *p* = 0.01) while using SBRT.

HCC with macrovascular invasion (MVI)—when the tumor extends into a vascular structure such as portal veins (PVTT), hepatic veins, or the inferior vena cava—tends to respond better to SBRT [16,62,63]. In this subset of HCC, TACE and TARE may not be effective, especially if the tumor invades larger veins [64,65,66]. Trials currently studying TARE and ICI combinations are excluding patients with MVI entirely (NCT05063565) or those with the invasion of large veins such as the inferior vena cava and main portal vein (NCT04541173; NCT04605731).

A meta-analysis published in 2018 (*n* = 2513 in 37 studies and 42 cohorts) compared three-dimensional conformal radiation therapy (3DCRT), TARE, and SBRT for HCC with PVTT [16]. Most studies were retrospective reviews, and response rates were available for 32/42 cohorts. The response rates and local control rates were higher with SBRT (71% and 89%, respectively) than TARE (33% and 58%) and 3DCRT (51% and 83%). The one- or two-year survival rates were not significantly different among these three treatment groups. The response rates and local control rates between TARE and SBRT were not directly compared here, but the response rates in 3DCRT were significantly higher and lower compared to TARE (*p* = 0.002) and SBRT (*p* = 0.001) groups, respectively, and local control rates with 3DCRT were significantly better than TARE (*p* = 0.001) and not significantly different than SBRT (*p* = 0.2) groups. SBRT showed similar response rates and local control rates in other studies [63,67]. The key takeaway is that higher response rates with SBRT did not provide a survival advantage—based on pre-clinical studies, this is something that can hopefully be achieved using ICI.

On the other hand, the addition of TKI to SBRT did not give a clear survival or response rate advantage in a similar population [39,63]. A retrospective study published in 2021 evaluated long-term outcomes in HCC (*n* = 128) with MVI treated with SBRT [63]. The median overall survival (OS) in the SBRT group was 18.3 months (*n* = 128). The patients who received sorafenib (*n* = 43, 14 after disease progression) following SBRT had a better OS (37.9 months). However, 14 patients who got TKI after disease progression had OS of just 7.5 months. A study from Taiwan showed that the addition of sorafenib did not significantly improve the survival rate even though the response rates were numerically better in the sorafenib group (81% vs. 74%) [39]. In 54 patient retrospective review published in 2020 (*n* = 54), patients with SBRT alone (*n* = 36) were compared to SBRT plus sorafenib (*n* = 18) showed no benefit of adding TKI [68]. The response rates and survival were not statistically different in both groups. Grade 3 adverse events such as rash, leukopenia and thrombocytopenia were common in TKI group while eleveated liver enzymes were reported in the SBRT-only group. Moreover, the toxicity in some early phase 1 trials was concerning, especially when high effective irradiated liver volume (veff) of 30% to 60% was used [40]. Adverse effects included bowel obstruction, bleeding, and worsening liver function. In recent years, SBRT plus sorafenib trials were terminated due to low accrual rates (NCT02989870 and NCT01005875). The combination of SBRT with VEGFi proved toxic in previous studies and should be explored with caution in any cancers [41,69]. On a side note, ramucirumab did not improve survival in MVI-positive HCC in the REACH-2 trial [64,70]. Post hoc analysis of MVI-positive HCC in the KEYNOTE 240 and CheckMate 459 trials (which also included patients with extra-hepatic disease) show the benefit of ICI in this population [19,64,71].

As noted in Table 2, only one trial explores SBRT plus ICI in this population (NCT04167293). Most of the studies of aHCCs have broad inclusion criteria, and it will be interesting to see the results for the MVI-positive HCC subset in the data. One trial adds TACE to SBRT and ICI (NCT04988945). Evidence from small older studies supports the TACE plus SBRT combination, but a new school of thought is to move away from using TACE in this group of HCCs [64,66,72,73]. In START-FIT trial, participants (*n* = 33 locally advanced HCC, 21 out of 33 had MVI) received SBRT after TACE, followed by avelumab (every 2 weeks) [73]. Participants with tumors invading larger veins (inferior vena cava or main portal vein) were excluded. The primary endpoint of downstaging to surgery was achieved in just 3 (9%) patients. The objective response rate, complete response and partial response reported were 63%, 44% (*n* = 15), 19% (*n* = 6), respectively. Ten patients had SAE, and 5 had immune-related adverse events.

In summary, responses to SBRT alone and ICI alone are promising in MVI-positive HCC, but neither of these individually provided a survival advantage. Combining them should be explored in this subset of HCC in large-scale trials.

## 4. SBRT Plus ICI in Cholangiocarcinoma

SBRT plus ICI combination therapy in CCA has limited evidence in the literature. Two case series reported the efficacy and safety of this combination in CCAs of all stages (details in Table 1) [57,58]. The reported response rates are promising even in patients who progressed on chemotherapy (paclitaxel/oxaliplatin, gemcitabine/oxaliplatin) and targeted therapy such as ERBB2-directed therapy (lapatinib). No significant toxicity was reported in both reports.

Two phase II clinical trials are underway to investigate the safety and efficacy of combined radiotherapy and ICI (Table 3). The CORRECT trial in China is recruiting patients with unresectable IHC (NCT03898895). It is an interesting, randomized trial (1:1) where SBRT plus ICI is tested in the first line against standard chemotherapy, GC. Participants will receive radiotherapy (either intensity-modulated radiation therapy (IMRT) or SBRT) followed by PD-1 blockade with camrelizumab. Patients will receive camrelizumab every three weeks until disease progression or unacceptable toxicity. The other group will receive GC for eight cycles. The primary outcome is two-year progression-free survival (PFS). Another smaller trial in China will evaluate the effectiveness of tislelizumab and radiotherapy (either IMRT or SBRT) in the second ine (post chemotherapy and no ICI) (NCT04866836). The patient will receive tislelizumab every three weeks three days after radiotherapy (IMRT or SBRT). Both trials were designed before the TOPAZ-1 trial era and new trials comparing SBRT plus ICI against GC/durvalumab or post-ICI are needed now. One phase I trial (NCT04708067) is combining hypofractionated radiation (15 fractions) and bintrafusp alfa, a bifunctional fusion protein targeting TGF-β and PD-L1, to treat aCCA in the second line, post-chemotherapy. Even though the evidence on the benefit of this approach is scarce for CCAs, given the options available in the current clinical practice, it is worth exploring.

### Identifying Ideal CCA Patients for SBRT Plus ICI

Previously, CCAs were considered radioresistant, but there is emerging evidence that radiation therapy can improve the outcomes [74]. Unlike HCC, there are no subsets of CCAs that respond better to SBRT. With the available evidence, it is not reasonable to propose the combination of SBRT plus ICI in the first line to any patient eligible for GC and durvalumab [75,76,77]. However, this approach can be an alternative for patients who cannot tolerate GC (secondary to poor performance status or debilitating neuropathy) and do not have any targetable mutations. On the other hand, the second-line chemotherapy options for these CCAs are not effective. FOLFOX improved OS by only one month (6.2 vs. 5.3 months, HR of 0.69, *p* = 0.031) compared to supportive care [24,78]. Nivolumab could be used in subsequent lines, and SBRT could be offered along with it if the patient did not receive durvalumab in the first line [28].

Even though earlier studies showed some benefit of combing chemotherapy with SBRT in the neoadjuvant setting before transplant and in locally advanced tumors, clinical trials (NCT01151761, NCT00983541) in recent years were terminated due to poor enrollment [36,37]. Very few current trials are exploring SBRT plus ICI (Table 3) in CCA compared to HCC. There is a need to focus on refining this approach to help patients with this deadly disease.

## 5. Future Directions

The evidence from retrospective reviews, case series, and phase I trials supports the efficacy and safety of SBRT plus ICI. Clinical trials are currently underway to investigate the long-term outcomes and mortality of this combination in larger populations (HCC > CCA). Patient selection for designing clinical trials for this approach is important, especially with ICIs moving to the first line in both types of tumors.

The subset of HCC with MVI (especially, main trunk, first- and second-order portal vein invasion) eligible for SBRT could be the starting point to test this combination, as TACE and TARE are not ideal for them [64]. Most aHCC patients have compromised liver function at diagnosis inhibiting the use of standard systemic options such as atezolizumab/bevacizumab or TKIs (based on the eligibility criteria of these trials). Rarely, patients move to the second line with reasonable liver function or performance status, allowing second/third line therapy use. This approach could introduce systemic therapy in early- or intermediate-stage tumors before compromising liver function. The HCCs with residual or recurrent disease after LRTs can benefit from this combination instead of repeating LRTs alone (if SBRT is feasible for them). SBRT plus ICI could be explored as a second line therapy (after disease progression on first-line ICI) in patients ineligible for TKIs. Future trials must be planned in these populations. Later, aHCCs with distant metastases and pre-surgery optimization of early-stage tumors can be studied. Gastrointestinal bleeding was reported in many studies and special attention to it is necessary while designing new trials.

SBRT plus ICI is a reasonable strategy for patients with unresectable CCAs (with preserved liver function and no targetable mutations) that progressed on GC (without durvalumab) or those who cannot tolerate GC. The success of the TOPAZ-1 trial brought durvalumab into the first line, but this combination could be explored as maintenance therapy or in second and third lines. Future trials should focus on these groups. Safety data is limited in these tumors and hence caution must be exercised during trial designs based on similar trials in other solid tumors.

## 6. Conclusions

Advanced primary liver tumors such as HCC and CCA have high mortality rates and very limited treatment options. SBRT plus ICI is a promising approach in these groups based on pre-clinical evidence and experiences with other solid tumors (lung and prostate). The goal is to make the response achieved by SBRT durable (in addition to enhancing anti-tumor immune activity) by adding ICI. There is preliminary evidence of the efficacy and safety for this combination in HCC and CCA, so it should be explored further in larger trials. Robust multicenter randomized controlled studies to identify the ideal populations and timing of SBRT relative to ICI (concurrent vs. sequential) are needed for this approach.

## Figures and Tables

**Figure 1 cancers-15-00050-f001:**
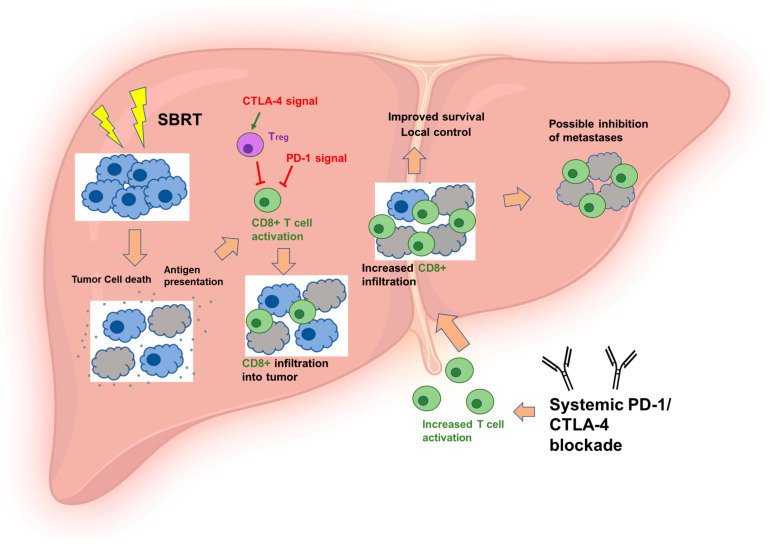
Synergistic effect of immunotherapy and SBRT on advanced liver tumors. SBRT—stereotactic body radiation therapy; PD-1—programmed death 1; CTLA-4—cytotoxic T-lymphocyte-associated antigen-4.

**Table 1 cancers-15-00050-t001:** Available Evidence on Stereotactic Body Radiation Therapy and Immune-Checkpoint Inhibitors in Hepatocellular Carcinoma and Cholangiocarcinoma.

Cancer	Study	Baseline Characteristics	ICI Agent Used	Response	Toxicity
Hepatocellular cancer	Case series (*n* = 5) [55]	4/5 had CP A 1/5 had CP B8	Nivolumab *	RECIST: PR in all patients (ranging from 30–84%) mRECIST: CR in 2/5 and PR in 3/5 1-yr OS: 100% None had PD **	≥grade 3 toxicity (both pneumonitis and skin) in one patient No cases of radiation-induced liver disease
		All had uHCC Single lesion Median size of tumors—9.8 cm (range: 9–16.1 cm) MVI positive—2/5 Distant metastatic disease—1/5 TACE—4/5			
	Phase I trial ^#^ (*n* = 14) [56]	CP A ^##^ Adequate organ function	Nivolumab (*n* = 6) Ipilimumab plus nivolumab (*n* = 8)	iRECIST: PR—36%; SD—36%; PD—28% Ipilimumab/nivolumab group: PR—50%; SD—37.5%; PD—12.5% Nivolumab group: PR—12.5%; SD—37.5%; PD—50%	DLT in 2 patients (hepatotoxicity)
		uHCC ineligible to resection or transplant ^###^ Can have multifocal or distant metastatic disease Can have TACE or ablation, but SBRT should be for a different lesion			
Cholangiocarcinoma	Case series (*n* = 3) [57]	Metastatic disease—1/3 Recurrent disease in 2/3 MSS, low TMB (0.98 to 3.8 muts/Mb) No systemic therapy in the first line	Pembrolizumab (*n* = 2) Nivolumab (*n* = 1)	CR in one PR in two (ranging from 41—86%)	No significant toxicity reported
	Case series (*n* = 4) [58]	One unresectable ^; 2 recurrent cases; one early stage Systemic therapy used in first-line include, paclitaxel/oxaliplatin, gemcitabine/oxaliplatin, and lapatinib	Pembrolizumab (*n* = 2) ^^ Nivolumab (*n* = 2)	Resection in unresectable SD in other 3 patients	No significant toxicity reported

CP—Child-Pugh Score; uHCC—unresectable hepatocellular carcinoma; MVI—macrovascular invasion; TACE—transcatheter arterial chemoembolization; RECIST- Response Evaluation Criteria in Solid Tumours; mRECIST—modified Response Evaluation Criteria in Solid Tumours; iRECIST—immune Response Evaluation Criteria in Solid Tumours; PD—progressive disease; OS—overall survival; DLY—dose-related toxicity; MSS—Microsatellite Stable; TMB—Tumor mutational burden; CR—complete response; PR—partial response; SD—stable disease. * in one patient, nivolumab was changed to pembrolizumab; ** at the time of publication; ^#^ trial was closed early: ^##^ eligibility criteria of the trial; ^ with multiple satellite lesions, regional lymphadenopathy, and extending to adrenal gland; ^^ one patient had pembrolizumab plus everolimus.

**Table 2 cancers-15-00050-t002:** Active Trials for Stereotactic Body Radiation Therapy and Immune-Checkpoint Inhibitors in Hepatocellular Carcinoma.

Identifier	Concurrent vs. Sequential	Phase	Investigating Arm	Comparative Arm	SBRT	Primary Outcome	Estimated Enrollment
NCT04167293	Sequential	III	SBRT followed in 4-6 weeks by IV sintilimab every 3 weeks ^#^	SBRT	30–54 Gy in 3–6 Fxs	PFS at 24 weeks	116
NCT04913480		II	Durvalumab every 2 weeks, 1 week prior to SBRT	N/A	27.5–50 Gy in 5 Fxs	PFS at 1 year	37
NCT04169399		II	Toripalimab every 3 weeks, followed by SBRT within 2 weeks	N/A	36–54 Gy in 6 Fxs	PFS in 6 months	30
NCT04988945		II	TACE + SBRT followed by durvalumab + tremelimumab ^#^	N/A	Not specified	Downstaging to resection	30
NCT03817736		II	TACE + SBRT followed by avelumab every 2 weeks ^R^	N/A	Not specified	Downstaging to resection	33 ^S^
NCT04193696	Concurrent	II	IMRT or SBRT, followed by carelizumab every 3 weeks	N/A	20–40 Gy in 10 Fxs	ORR at 6 months	39
NCT03857815		II	SBRT + sintilimab every 3 weeks^R^	N/A	Not specified *	PFS at 2 years	30
NCT04547452		II	SBRT + sintilimab every 3 weeks	IV sintilimab every 3 weeks	35–80 Gy in 5–8 Fxs *	PFS at 24 weeks	84
NCT03316872		II	Pembrolizumab every 3 weeks, with SBRT starting day 2	N/A	5 Fxs over 8–15 days **	ORR	30
NCT03482102		II	Durvalumab + tremelimumab every 28 days, radiation during cycle 2 ***	N/A	Not specified	ORR	70
NCT05185531 Neo-adjuvant		I	SBRT and tislelizumab on days 1 and 22, followed by tumor resection	N/A	8 Gy in 3 Fxs	Delay to surgery	20
Terminated trials University of Chicago: Phase I study on SBRT + Nivolumab/Ipilimumab (NCT03203304)—poor accrual							

SBRT—stereotactic body radiation therapy; Fxs—fractions; PFS—progression-free survival; ORR—objective response rate; N/A: not applicable as they are single-arm studies; * radiation can be given to liver or lung or any metastatic lesion; ^#^ specific for patients with portal vein tumor thrombosis; ** in accordance with institutional protocol; *** Not specifically SBRT; ^R^—early results available; ^S^—stopped recruiting.

**Table 3 cancers-15-00050-t003:** Active Trials for Stereotactic Body Radiation Therapy and Immune-Checkpoint Inhibitors in Cholangiocarcinoma.

Identifier	Phase	Investigating Arm	Comparative Arm	Radiation Dose	Primary Outcome	Target Accrual
NCT03898895	II	IMRT/SBRT followed in 1 week by camrelizumab every 3 weeks	GC	40 Gy as SBRT or IMRT	2-year PFS	184 (1:1)
NCT04866836	II	IMRT/SBRT followed in 3 days by tislelizumab every 3 weeks *	N/A		ORR in 2 years	20
NCT04708067	I	Hypofractionated radiation followed by bintrafusp alfa every 2 weeks *	N/A	15 fractions	Incidence of adverse events	15
Terminated trials American University of Beirut Medical Center: Phase II pilot study on SBRT + nivolumab after induction chemotherapy in cholangiocarcinoma (NCT04648319) due to poor accrual						

SBRT—stereotactic body radiation therapy; IMRT—intensity-modulated radiation therapy; N/A- not applicable as they are single-arm studies; GC—gemcitabine + cisplatin; PFS—progression-free survival; ORR—objective response rate; * second-line post chemotherapy.

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
