# Peer review of "Stereotactic Body Radiation Therapy (SBRT) Plus Immune Checkpoint Inhibitors (ICI) in Hepatocellular Carcinoma and Cholangiocarcinoma"

_cancers, 2022, doi:10.3390/cancers15010050_

Round 1

Reviewer 1 Report

Dear Authors,

This paper investigates the usefulness of the combination of SBRTx and ICI in patients with advanced HCC and cholangiocarcinoma. I think it is quite impressive and readers will be interested in this well-organized paper.

However, I think there are some points that need improvement and/or correction in your paper. 

I ask for appropriate responses after read the followings.

1. Please provide more information about advantages of SBRT over conventional RTx. 

Is there any previous report that SBRT can reduce adverse events in radiosensitive organs around the liver? (such as pneumonitis, duodenitis, or peptic ulcer..)

2. Cholangiocarcinoma is known to be a radioresistant cancer, and there are reports that RT induces lymphocyte depletion. Can you give supplement comments to this point ?

3. Please comment on whether there are predictive markers for this combination (SBRT + ICI), such as PD-L1 expression, MSI-H, and TMB, or whether SBRT + ICI is beneficial regardless of markers.

Author Response

We appreciate the time and effort of the reviewer. Please find the responses to the specific questions below. 

  1. Please provide more information about advantages of SBRT over conventional RTx. Is there any previous report that SBRT can reduce adverse events in radiosensitive organs around the liver? (such as pneumonitis, duodenitis, or peptic ulcer..)
    • We appreciate the great feedback. We added few lines in the introduction addressing this as follow, Severe adverse events such as duodenal ulcer, pneumonitis, and gastrointestinal bleeding were associated with traditional (external beam) radiation but not with SBRT according to the metanalysis that compared various radiation modalities in HCC patients with portal vein tumor thrombus (PVTT)[16]. While lymphocytopenia is the most common adverse event with traditional radiation, severe thrombocytopenia was the most frequently reported adverse event (15%). Resection is offered to eligible early-stage tumors whenever it is feasible.
  2. Cholangiocarcinoma is known to be a radioresistant cancer, and there are reports that RT induces lymphocyte depletion. Can you give supplement comments to this point ?
    • We appreciate the great feedback. Cholangiocarcinoma was considered radioresistant in the past, but there is reasonable evidence that it will improve the outcomes in recent years. Lymphocytopenia from radiation is not specific to cholangiocarcinoma. To this point, we added a line in section 4.1 with references that address this very concern, Previously, CCAs were considered radioresistant, but there is emerging evidence that radiation therapy can improve the outcomes [75].

  1. Please comment on whether there are predictive markers for this combination (SBRT + ICI), such as PD-L1 expression, MSI-H, and TMB, or whether SBRT + ICI is beneficial regardless of markers.
    • We appreciate the great feedback. There are no reliable predictors for response to this combination to our knowledge. We acknowledged it in section 3.1, by adding a line, To our knowledge, there are no reliable predictors for a robust response for SBRT plus ICI.

Reviewer 2 Report

This reviewer was excited upon the title of the manuscript "SBRT plus ICI in HCC and CCA", but was quickly disappointed since there was little data on either HCC or CCA both clinically and pre-clinically. Therefore, the title of the manuscript needs to changed to "Lack of evidence (data) regarding benefit(s) of SBRT plus ICI in HCC or CCA" or something like that.

Overall this is a good review with many useful information. But, on the very subject, it begs data from both clinical and preclinical studies.  When the authors insisted this combo was "promising", they based their believe only on other cancers such as the lung cancers... There are other issues such as the relationship between response and OS, which somehow diluted the story line. Maybe the best way is to modify the title without extensive editing since some of the trials are still ongoing, and other new trial ideas were just proposed...

Author Response

We appreciate the great feedback. We intentionally chose a neutral/unbiased title for the review reflecting the available evidence. We want readers to get the most accurate information and make their own opinion on this topic. We understand that we cannot extrapolate the evidence/data from lung cancers, and hence cautiously used the language expressing our hope (but not definitive statements) based on the literature in the review. Unfortunately, we have limited data on this topic, and we hope to initiate the appropriate discussion as we are desperate to improve the outcomes of these cancers.

Reviewer 3 Report

Cancers Review:

In simple summary would remove "(tube system on the liver responsible for 18 pushing bilirubin into the gastrointestinal tract) " this line is too simplistic. 

In simple summary would rephrase: "The recent success of immunotherapy where in patient’s immune system is activated to fight the tumor is encouraging  but only a fraction of liver cancer eligible patients is for it." to. "The recent  success of immunotherapy where in patient’s immune system is activated to fight the tumor is encouraging but on a fraction of patients with liver cancer remain eligible for this treatment"

In abstract - why should those patients that are ineligible for Bev be considered - there are many trials looking at SBRT in addition to Atezo/Bev therapy

In intro - surgery is not considered a loco regional treatment.   I would not say that pembro and nivo can't be considered in patients prior on ICI - would recommend you just say these are second line agents or for those who progressed on first line therapy 

In simple summary would rephrase: "The recent success of immunotherapy where in patient’s immune system is activated to fight the tumor is encouraging  but only a fraction of liver cancer eligible patients is for it." to. "The recent  success of immunotherapy where in patient’s immune system is activated to fight the tumor is encouraging but on a fraction of patients with liver cancer remain eligible for this treatment"

In abstract - why should those patients that are ineligible for Bev be considered - there are many trials looking at SBRT in addition to Atezo/Bev therapy

In intro - surgery is not considered a loco regional treatment.   I would not say that pembro and nivo can't be considered in patients prior on ICI - would recommend you just say these are second line agents or for those who progressed on first line therapy 

On page 7 would be helpful to define the classes each of the ICIs are in that are mention in current trials (ex PD1 or CTLA4 inhibitors)

You describe the mechanism graphically in which radiation + ICI may provide synergistic effects.  I would touch upon the STAR effect and would give a background on the theory behind TKI or VEGFi with radiation and how this may be different than with ICI for which we hope to see more promising outcomes

In conclusion you note that TARE is not ideal for patients with vascular invasion, however this is actually an ideal population given the lack of embolic effect on the hepatic artery for which this form of locoregional therapy is preferred (as long as not main PV).

Author Response

We appreciate the time and effort of the reviewer. Please find the responses to the specific questions below. 

  1. In simple summary would remove "(tube system on the liver responsible for 18 pushing bilirubin into the gastrointestinal tract) " this line is too simplistic. 
    • We appreciate the feedback. The goal of this line is to explain the biliary tract to non-medical personnel. We will remove it if it comes across as too simplistic.

  1. In simple summary would rephrase: "The recent success of immunotherapy where in patient’s immune system is activated to fight the tumor is encouraging  but only a fraction of liver cancer eligible patients is for it." to. "The recent  success of immunotherapy where in patient’s immune system is activated to fight the tumor is encouraging but on a fraction of patients with liver cancer remain eligible for this treatment"
    • We appreciate the feedback. We changed it as suggested. Now that line reads, The recent success of immunotherapy where in patient’s immune system is activated to fight the tumor is encouraging, but only a fraction of patients with liver cancer remain eligible for this treatment

  1. In abstract - why should those patients that are ineligible for Bev be considered - there are many trials looking at SBRT in addition to Atezo/Bev therapy
    • We appreciate the feedback. We intended to propose the combination of SBRT plus ICI in patients who do not qualify for the standard of care, such as bevacizumab and TKI based on available evidence. We acknowledge that trials are combing SBRT with bevacizumab but also recognize the concern of bowel toxicity associated with such combinations (ref-41).

  1. In intro - surgery is not considered a locoregional treatment.   I would not say that pembro and nivo can't be considered in patients prior on ICI - would recommend you say these are second line agents or for those who progressed on first line therapy 
    • We appreciate the feedback. We removed surgery from line 57 and added an additional line Resection is offered to eligible early-stage tumors. To avoid confusion, we deleted, who did not receive ICIs initially from line 73.

  1. On page 7 would be helpful to define the classes each of the ICIs are in that are mention in current trials (ex PD1 or CTLA4 inhibitors)
    • We appreciate the feedback. We added the information at the advised place in the manuscript.

  1. You describe the mechanism graphically in which radiation + ICI may provide synergistic effects.  I would touch upon the STAR effect and would give a background on the theory behind TKI or VEGFi with radiation and how this may be different than with ICI for which we hope to see more promising outcomes
    • We appreciate the feedback. As the paper’s focus was specifically on SBRT plus ICI, we did not discuss the combination of TKI or VEGFi in the figure. We the following lines to the paper in section 2, ICIs that target PD-1 increase the polarization of tumor-associated macrophages and activate cytotoxic T cells. Anti-PD-L1 inhibits tumor immune evasion. Anti-CTLA4 also inhibits immune evasion and increases interaction between dendritic cells and T cells. Combining ICIs with VEGFis or TKIs enhance these effects by promoting dendritic cell and T lymphocyte activity and inhibiting regulatory T cells and myeloid-derived suppressor cells. This results in increased anti-tumor inflammation and a more durable response to ICIs

  1. In conclusion you note that TARE is not ideal for patients with vascular invasion, however this is actually an ideal population given the lack of embolic effect on the hepatic artery for which this form of locoregional therapy is preferred (as long as not main PV).
    • We appreciate the feedback. We acknowledge TARE is not ideal for larger vein invasion disease. We could not figure out which specific line in the manuscript this comment was directed at, but we clarified this in section 5, line 341.